# Pharmaco-Epidemiological Study and Correlation Between Antibiotic Resistance and Antibiotic Consumption in a Tunisian Teaching Hospital from 2010 to 2022

**DOI:** 10.3390/antibiotics14020135

**Published:** 2025-02-01

**Authors:** Yosr Kasbi, Fatma Sellami, Asma Ferjani, Aimen Abbassi, Ilhem Boutiba Ben Boubaker

**Affiliations:** 1Internal Pharmacy Service, Charles Nicolle Hospital, Tunis 1007, Tunisia; abbassi_aimen@yahoo.fr; 2Faculty of Pharmacy of Monastir, University of Monastir, Monastir 5000, Tunisia; 3Laboratory of Microbiology, Charles Nicolle Hospital, Tunis 1006, Tunisia; aferjani76@gmail.com (A.F.); ilhem.boutiba@rns.tn (I.B.B.B.); 4Research Laboratory “Antimicrobial Resistance”, LR99ES09, Faculty of Medicine of Tunis, University of Tunis El Manar, Tunis 1007, Tunisia

**Keywords:** Tunisian hospital, antibiotics, antimicrobial resistance, AWaRe classification, correlation, consumption

## Abstract

The exponential rise of bacterial resistance poses a threat to antibiotic efficacy, with a great impact on public health. This study aims to analyze the correlation between antibiotic consumption and the emergence of bacterial resistance. Conducted retrospectively at Charles Nicolle Hospital in Tunis, Tunisia, from 2010 to 2022, this study was based on STKMED^®^ software for antibiotic consumption data, hospital administrative records for the number of hospitalization days, and SIRSCAN^®^ software for bacteriological data. Data processing was performed using Excel^®^ software version 2019, and analysis was conducted using SPSS23^®^. In 2022, consumption was almost evenly split between the two major “AWaRe” groups, with 49.33% for “Access” and 46.89% for “Watch”, and the consumption of the “Reserve” group also increased, accounting for 3.77% of the total. Bacterial resistances, notably carbapenem-resistant *Klebsiella pneumoniae*, increased. Seventy-four significant correlations were identified, including those between carbapenem consumption and resistance in *Escherichia coli* and *Klebsiella pneumoniae* strains. However, no significant correlation was observed with imipenem-resistant *Pseudomonas aeruginosa* strains. The significant correlations between the emergence of bacterial resistance and antibiotic use, particularly with antibiotics in the “Watch” and “Reserve” groups, underscore the urgent need to continue efforts to combat this threat through rational antibiotic use.

## 1. Introduction

The discovery of antibiotics in the early 1930s was the miraculous solution that radically changed the management of various infections, whether community-acquired or nosocomial, and saved thousands of lives [1]. Unfortunately, currently, the effectiveness and reliability of these antimicrobial agents are compromised due to the multitude of bacterial resistances that have begun to emerge for several decades, shortly after the beginning of antibiotic use. This evolution makes the choice of therapeutic protocol more complex and plunges the world into a critical postantibiotic phase, with an increase in morbidities and mortalities [2,3,4]. In 2021, the World Health Organization (WHO) classified antibiotic resistance as one of the top 10 threats to public health [5]. The increasing numbers, with 2,868,700 infections due to resistant bacteria and 35,900 deaths due to antibiotic-resistant bacterial infections annually, confirm this observation [5,6,7]. Hamad et al. estimate that if this problem is not addressed promptly, it could kill up to 10 million people annually by 2050 and incur expenses exceeding 100 trillion US dollars [2]. Naturally, bacteria may have inherent resistance to certain antibiotics, but these resistances can be amplified by the acquisition of additional resistance genes often driven by human activities such as the inappropriate use of antibiotics [7]. WHO advocates that only 50% of antibiotics are used correctly [8]. It has been approved that the use, misuse, and overuse of antibiotics have led to direct and indirect correlations with the emergence of bacterial resistance through the phenomenon of selection pressure from resistant mutants [9,10].

Several studies have focused on the evolution of antibiotic consumption and others on bacterial resistances prevailing in Tunisian hospitals; however, the correlation between these two parameters has rarely been addressed in Tunisia, and it has never been studied at our hospital or on a large scale in the Greater Tunis area, especially over an extended period.

In this context, the main objective of this study is to analyze the correlation between antibiotic consumption and the emergence of bacterial resistance at Charles Nicolle Hospital (CNH) in Tunis over a 13-year period.

## 2. Results

### 2.1. Antibiotic Consumption

#### 2.1.1. The “AWaRe” Classification (Table A1)

Antibiotic consumption in the “Access” group of the “AWaRe” classification was 610.84 DDD/1000 PD in 2010 but significantly decreased to 185.15 DDD/1000 PD by 2022.

For the “Watch” group, the variation in consumption between 2010 and 2022 was minimal, with the lowest consumption recorded at 173.75 DDD/1000 PD.

The consumption of antibiotics in the “Reserve” group fluctuated over the years. In 2010, it was only 5.3 DDD/1000 PD, gradually increasing to 16.25 DDD/1000 PD in 2015. A drop occurred in 2019 with 6.15 DDD/1000 PD, followed by a resurgence, reaching 14.17 DDD/1000 PD in 2022 (Figure 1).

In 2010, the majority of antibiotic consumption was represented by the “Access” group, with 73.17%, compared to only 26.19% for the “Watch” group and 0.64% for the “Reserve” group. Over the years, this distribution has changed significantly, with a considerable decrease in the share of the “Access” group and an increase in those of the “Watch” and “Reserve” groups. By 2022, consumption was almost evenly split between the two major groups, with 49.33% for “Access” and 46.89% for “Watch”, and the consumption of the “Reserve” group also evolved, accounting for 3.77% (Figure 2).

#### 2.1.2. The Linear Regression of Antibiotic Consumption

The linear regression of overall consumption in DDD/1000PD is significant (P = 0.000; β = −0.837), indicating a significant downward trend.

Table 1 shows the results of the linear regressions of the studied antibiotic consumption.

### 2.2. Bacterial Resistance (Figure 3)

#### 2.2.1. Enterobacteriaceae Resistant to Third-Generation Cephalosporins

From 2010 to 2015, the resistance of *E. coli* strains to third-generation cephalosporins showed an upward trend, rising from 8.4% to 23%. However, from 2016 to 2022, this resistance gradually decreased, reaching 14%.

The evolution of resistance in *K. pneumoniae* strains to third-generation cephalosporins has fluctuated over the years, with a minimum of 32% recorded in 2015 and a maximum of 45% observed in both 2019 and 2022.
Figure 3Overlay of the evolution of resistance among the different bacterial strains studied. R: resistance; *P.: Pseudomonas; K.: Klebsiella; A.: Acinetobacter; E.: Escherichia*; MRSA: methicillin-resistant *S. aureus*.
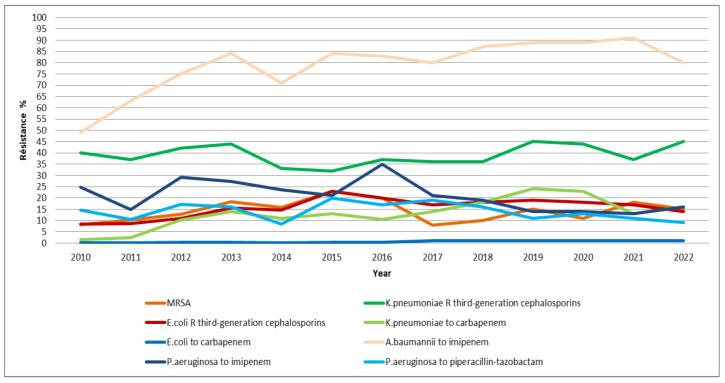


#### 2.2.2. Enterobacterial-Producing Extended-Spectrum Beta-Lactamase

The level of ESBL-producing *E. coli* steadily increased from 7.91% in 2011 to 20.68% in 2016, then began to decline, reaching its lowest value of 7.16% in 2020.

The percentage of ESBL-producing *K. pneumoniae* steadily decreased from 37% in 2012 to 12% in 2020. However, there was a resurgence in 2021, with the percentage rising to 25.96%, and it further increased to 28.13% in 2022.

#### 2.2.3. Enterobacterial Resistance to Carbapenems

This study showed a low resistance (<1%) of *E. coli* strains to carbapenems.

The resistance of *K. pneumoniae* strains to carbapenems saw a sharp increase, rising from 1.42% in 2010 to 14.02% in 2013. A second peak occurred in 2019 at 24% and finally settled at 16% in 2022.

#### 2.2.4. *Acinetobacter baumannii* Resistant to Imipenem and Piperacillin-Tazobactam

The resistance of *A. baumannii* to imipenem has significantly evolved, increasing from 49% in 2010 to 80% in 2022, reaching a peak of 91% recorded in 2021.

#### 2.2.5. *Pseudomonas aeruginosa* Resistant to Imipenem and Piperacillin-Tazobactam

Between 2010 and 2016, the resistance of *P. aeruginosa* to imipenem fluctuated, with a low of 14.78% in 2011 and a high of 35% in 2016. Thereafter, this resistance gradually decreased, reaching 16% in 2022, with a minimum of 13% observed in 2021.

The resistance of *P. aeruginosa* to piperacillin-tazobactam PTZ has fluctuated over time. A minimum percentage of 8.48% was observed in 2014, followed by another low of 9% in 2022, while a peak of 20% was recorded in 2015.

#### 2.2.6. Methicillin-Resistant *Staphylococcus aureus* Strains

There is no overall increasing trend in methicillin resistance among *S. aureus* (MRSA). Minor variations, particularly noted from 2019 onwards, show a minimum resistance of 8% in 2017 and a peak resistance recorded in 2015 of 23%.

#### 2.2.7. Linear Regressions of Bacterial Strain Resistance Densities

Table 2 presents the results of linear regressions of resistance densities of the studied bacterial strains to antibiotics. Four significant regressions were identified.

### 2.3. Correlations Between Antibiotic Consumption and Bacterial Resistance Incidence Density

The correlation study between antibiotic consumption and the incidence density of bacterial resistance was conducted for each bacterial–antibiotic pair. Prior to this, the Shapiro–Wilk test was performed to differentiate between variables with a normal distribution and those with a nonnormal distribution. Subsequently, the Spearman coefficient was used to assess correlation when both tested variables were nonnormally distributed, while the Pearson coefficient was utilized when at least one of the variables followed a normal distribution. The direction of a correlation is determined only if the regression of antibiotic consumption is significant (*p*-value ≤ 0.05).

Significant correlations found with *E. coli* strains are presented in Table 3, Table 4 and Table 5.

Significant correlations found with *K. pneumoniae* strains are represented in Table 6 and Table 7.

Significant correlations found with *Pseudomonas aeruginosa* strains resistant to piperacillin-tazobactam are represented in Table 8.

Significant correlations found with imipenem-resistant *Acinetobacter baumannii* strains are represented in Table 9.

Significant correlations found with Methicillin-resistant *Staphylococcus aureus* (MRSA) strains are represented in Table 10.

In total, 74 significant correlations were found. Among the most relevant are the correlations between carbapenem consumption and the increase in resistance density of *E. coli* strains to third-generation cephalosporins (C3G) and carbapenems, and of *K. pneumoniae* strains to carbapenems.

Seven of the 74 correlations had an undefined direction since the regression of the antibiotic consumption in question was not significant;

Therefore, while a correlation exists, its trend cannot be defined. It is noteworthy that no significant correlation was found between antibiotic consumption and the increase in resistance density of ESBL-producing *K. pneumoniae* strains or imipenem-resistant *P. aeruginosa* strains.

## 3. Discussion

Four million nine hundred fifty thousand (4.95 million) deaths were estimated worldwide in 2019 due to antimicrobial resistance (AMR), with 1.27 million directly attributable to bacterial AMR [11,12]. A close correlation has been revealed between the irrational use of antibiotics and the occurrence of these uncontrollable resistances, particularly caused by hospital pathogens [7,13].

Given the absence of previous studies conducted in hospitals in Greater Tunis, particularly at our hospital, which is the largest in terms of the number of beds in Tunisia, on this issue, we decided to reveal all correlations between antibiotic consumption and bacterial resistance.

The linear regression of overall antibiotic consumption was significant, showing a marked downward trend. Similarly, in India, consumption decreased by 3.6% between 2011 and 2019 [14]. A slight deviation from this trend was observed during the COVID-19 period. Indeed, comparing the consumption in 2019 (399.19 DDD/1000 PD) to that in 2020 (471.92 DDD/1000 PD), we can see an 18.22% increase in consumption due to the pandemic. According to the review by Fukushige et al., studies conducted in hospitals in Lebanon, Italy, India, Spain, and the United Kingdom revealed an increase in antibiotic consumption in 2020 compared to 2019 [15].

Overall, a considerable decrease in the consumption of antibiotics classified under the “Access” group, according to the “AWaRe” classification, was noted. In 2010, this group accounted for the highest share of antibiotic consumption at 73.17%, compared to only 49.33% in 2022. This percentage is well below the target of the WHO’s 13th General Programme of Work, spanning from 2019 to 2023, which aimed for at least 60% of total antibiotic consumption to be comprised of “Access” group drugs [16,17,18]. This decrease in consumption is probably due to changes in medical prescriptions and increased consumption of molecules belonging to the other “AWaRe” groups. Our results align with those reported in the 2019 report by the French National Agency for Medicines and Health Products Safety (ANSM), where the “Access” group represented 72% of total antibiotic consumption in France [19].

Antibiotics in the “Watch” group accounted for only 26.19% of consumption in 2010, but this percentage significantly increased in 2022 to 46.89%, nearly matching the consumption of the “Access” group. In a study conducted in the urology department of our hospital over the period from 2015 to 2019, it was revealed that the “Watch” group even surpassed the “Access” group with 60% of consumption compared to only 35% [20]. This percentage is alarming and absolutely discordant with WHO recommendations. This consumption must be reduced, notably by promoting therapeutic de-escalation and transitioning to “Access” group medications whenever possible. Cephalosporins, particularly cefotaxime, are the most consumed molecules in this group. This is also the case in China, according to a study conducted in 153 hospitals [21].

For the “Reserve” group, consumption has fluctuated over time. Antimicrobial use in this group is tightly controlled to preserve its effectiveness and reliability. A study conducted in the urology department of our hospital demonstrated that antimicrobial stewardship measures had significantly reduced the consumption of the “Reserve” group [20].

Regarding the resistance profile, our study identified a significant upward trend in the incidence density of *E. coli* resistance to third-generation cephalosporins (3GC). These findings are very close to those published by the Antibiotic Resistance in Tunisia LART [22]. In France, *E. coli* resistance to 3GC increased from 7% in 2010 to 8.3% in 2021 [23,24].

The results showed that the resistance density of *K. pneumoniae* to 3GC was highly variable, resulting in a nonsignificant linear regression of the distribution. These figures are consistent with those reported by LART [22]. The average percentage of *K. pneumoniae* resistant to 3GC in European Union countries recorded in 2021 was 34.3%, more or less comparable to that found in this study [25].

An increase in the percentage of *E. coli* and *K. pneumoniae* ESBL was highlighted in this study. In Australia, the recorded percentages in 2018 were significantly lower than those in our study for *K. pneumoniae* (9.8%) but were comparable for *E. coli* (13.3%) [26].

A significant upward trend in the resistance density of *K. pneumoniae* to carbapenems was recorded, with the percentage increasing from 1.42% in 2010 to 16% in 2022. Alarmingly high levels of carbapenem-resistant *K. pneumoniae* strains have been reported worldwide, with an upward trend [27]. This trend is justified by the recently acquired ability of *K. pneumoniae* to develop resistance genes, particularly against carbapenems [28].

The resistance of *P. aeruginosa* to imipenem and PTZ was marked by inconsistent trends, resulting in statistically nonsignificant evolution. For imipenem resistance, in a Tunisian burn intensive care unit, the average resistance was 74.9% between 2014 and 2019, higher than the result found in our study. This large difference in resistance rates is undoubtedly due to the increased involvement of *P. aeruginosa* in infections developed in burn patients [29].

High levels of resistance of *A. baumannii* to imipenem have been demonstrated. These resistances increased from 49% in 2010 to 80% in 2022. The development of carbapenemases has made *A. baumannii* “an even more dangerous threat,” as characterized by Ramirez et al. in their review [30]. A study conducted in 9 hospitals in Greece revealed resistance rates ranging from 90.3% in 2010 to 94.5% in 2015 [31].

The variations in MRSA were minimal during the study period, with extreme percentages of 8% in 2017 and 23% in 2015. These results are close to those presented by LART, with extreme figures of 15.2% in 2010 and 2022 and 23.4% in 2013 [22]. A study conducted in Iran between 2010 and 2016 revealed an MRSA rate averaging 43%, almost double the result we found [32]. This difference in resistance rates between our study and the one conducted in Iran can be justified by differences in healthcare systems, antibiotic consumption, antibiotic use protocols, and hygiene measures adopted in healthcare settings [33].

Regarding the relationship between antibiotic consumption and the emergence of bacterial resistance, 74 correlations have been revealed.

Resistance of *E. coli* to 3GC has been associated with the consumption of several antibiotics, particularly 3GC, such as cefotaxime. Chinese and Korean studies have confirmed these correlations [34,35]. However, unlike the result found in our study, in Italy, no link was found with the use of piperacillin-tazobactam [36].

Furthermore, increased consumption of carbapenems, notably imipenem, has been correlated with increased resistance of *E. coli* to 3GC, as was the case in Korea [37]. Conversely, a decrease in the consumption of fluoroquinolones, such as ciprofloxacin and ofloxacin, highlighting the importance of controlling the use of these antibiotics to reduce resistance rates, has been negatively associated with this correlation [37]. Aminoglycosides have also shown a significant correlation with *E. coli* resistance to 3GC, as indicated in an Italian study [38]. Likewise, a decrease in the consumption of cotrimoxazole has been associated with an increase in this resistance, according to the same Italian study [38].

A significant negative correlation has been observed between the consumption of penicillins, particularly oxacillin and amoxicillin, and the occurrence of ESBL-producing *E. coli*. In China, Yang P et al. demonstrated this correlation [39]. Similarly, significant correlations, positive with imipenem (and consequently with carbapenems) and negative with fluoroquinolones, notably ciprofloxacin, and with cotrimoxazole, have been revealed concerning the emergence of ESBL-producing *E. coli*. Similar correlations have been revealed in a Moroccan study (except for imipenem) [40].

A significant increase in the consumption of carbapenems, particularly imipenem, has led to a notable correlation with the resistance density of *E. coli* to this class of antibiotics. Conversely, the decreased consumption of ertapenem has resulted in a significant negative correlation. A study conducted in a Chinese hospital between 2010 and 2016 highlighted the correlation between imipenem consumption and the emergence of this resistance [41]. For ertapenem, similar findings were reported in a Serbian study [13].

Additionally, a significant negative correlation has been observed between the consumption of fluoroquinolones and the resistance density of *E. coli* strains to carbapenems. This correlation was documented in a Thai study conducted between 2013 and 2016 [42].

The increased consumption of 3GCs, particularly cefotaxime, has been correlated with the rise in resistance density of *K. pneumoniae* to 3GCs. Studies conducted in China between 2014 and 2016, as well as at the National Institute of Hygiene in Lomé between 2010 and 2017, have revealed these correlations [43,44].

The high consumption of imipenem in our hospital has been correlated with the emergence of carbapenem-resistant *K. pneumoniae*, as demonstrated by Pérez-Lazo et al. in a study conducted in a Peruvian hospital between 2015 and 2018 [45]. Conversely, the reduced consumption of certain penicillins (oxacillin, amoxicillin, and piperacillin) and fluoroquinolones has shown significant negative correlations with resistance. This is supported by a Sicilian study of the entire beta-lactam family [46] and a Thai study for fluoroquinolones [42].

The increased consumption of PTZ, 3GC, and teicoplanin has been positively correlated with the resistance density of *K. pneumoniae* to carbapenems. For PTZ, the correlation was not significant in Peru [45], while for 3GCs, particularly ceftriaxone and teicoplanin, these correlations were demonstrated in Thailand and China, respectively [42,47].

The consumption of the entire aminoglycoside family, particularly gentamicin, has shown correlations of an undefined nature with the resistance density of *P. aeruginosa* to PTZ. This finding was also observed in Serbia within an intensive care unit between 2014 and 2018 [48]. In our study, a significant positive correlation was recorded between the increased consumption of imipenem and the emergence of resistance in *A. baumannii* to this molecule. Studies conducted in China, Jordan, and Sahloul Hospital in Sousse have also noted the significance of this correlation [33,47,49].

Indirect correlations with this resistance have been recorded: positive correlations with PTZ and 3GC, demonstrated respectively in Spain in 2019 [50] and in Italy [38], and negative correlations with amoxicillin, as revealed in China [51], fluoroquinolones, as shown in a 15-year Chinese study and between 2015 and 2018 in Peru [45,52], and rifampicin. For rifampicin, its synergistic action with colistin in treating infections caused by multidrug-resistant *A. baumannii*, particularly those resistant to carbapenems, has produced favorable results [53]. However, the increased consumption of rifampicin has begun to impact the emergence of resistance in *A. baumannii* to carbapenems.

The consumption of oxacillin has shown a significant negative direct correlation with the occurrence of MRSA. Mascarello, M. et al. highlighted this negative correlation [36].

Regarding indirect correlations with MRSA, PTZ, and 3GC, particularly ceftriaxone, imipenem, and aminoglycosides, have been implicated in significant positive correlations. These correlations have been demonstrated in Italian [36] and Chinese studies [43,47,54].

Other correlations revealed in our study were not statistically significant in the literature. For example, the correlation between the consumption of 3GCs and the emergence of carbapenem-resistant *E. coli* [21] and the correlation between vancomycin and ESBL-producing *E. coli* [55].

This pharmaco-epidemiological study underscores the critical importance of bolstering antibiotic stewardship programs to optimize antibiotic prescriptions and prevent the unfavorable progression of bacterial resistance, which could lead to a therapeutic dead end.


Strengths of the study:



This study is the first of its kind conducted at Charles Nicolle Hospital in Tunis and the first across all hospitals in Greater Tunis to focus on the correlation between antibiotic consumption and the emergence of resistant bacterial strains.The extensive period covered by this study has provided significant insights into the evolution of antibiotic use and bacterial resistance, which were subsequently analyzed for correlation.To test these correlations, we employed both parametric and nonparametric methods, yielding more robust results.



Limitations of the study:


However, this study had several limitations:
It is a single-center study that only involves one hospital.It is a retrospective study and not a prospective one.Additionally, we excluded other factors that could influence bacterial resistance, such as patients’ clinical status (including age, weight, nutritional status, and medical, surgical, and family history), as well as hygiene conditions in clinical departments.

## 4. Materials and Methods

A descriptive retrospective study was conducted at Charles Nicolle Hospital in Tunis. Charles Nicolle Hospital is a public university hospital center, founded in 1897 under the French protectorate. It currently has a capacity of 950 beds and 30 cribs. The distribution into departments based on specialties is in pavilions, more or less independent for each specialty.

This study included:➢Systemic antibiotics administered orally or intravenously for curative or prophylactic purposes are prescribed in all clinical departments of the hospital. These antibiotics are classified according to the WHO’s “AWaRe” classification (A for “Access”, Wa for “Watch” and Re for “Reserve”) with their respective ATC (Anatomical Therapeutic Chemical Classification System) codes [56,57]. The objective of this classification is to define a standardized framework for the effective use of antimicrobials [19] in order to rationalize their use globally and curb antimicrobial resistance. In this context, an “AWaRe” book has been written [58], providing detailed guidelines to follow when prescribing antibiotics. This book addresses 34 infections and serves as a guide, especially for the empirical indications of antibiotics [16].The “Access” group includes antibiotics recommended as first- and second-line treatments in protocols for common infections. The “Watch” group also includes first- and second-line antibiotics but for specific infections rather than common ones. Since the drugs in this group have a higher potential for developing resistance, they must be an integral part of antimicrobial stewardship programs ASP. The “Reserve” group includes last-resort antibiotics used when the patient’s life is at risk due to an infection caused by multidrug-resistant bacteria [11,59,60].➢Nonredundant bacterial strains isolated from positive cultures obtained from various types of samples received from hospitalized patients at the hospital from 1 January 2010 to 31 December 2022 and analyzed in the microbiology laboratory were considered. The bacterial strains selected for study were chosen among the main bacteria posing a problem of antimicrobial resistance at our hospital and nationally: Methicillin-resistant *Staphylococcus aureus* (MRSA) strains, *Klebsiella pneumoniae (K. pneumoniae)* and *Escherichia coli (E. coli)* strains resistant to third-generation cephalosporins (3GC) with or without extended-spectrum beta-lactamase (ESBL) production, Carbapenem-resistant *Klebsiella pneumoniae (K. pneumoniae)* and *Escherichia coli (E. coli)* strains, Imipenem-resistant *Acinetobacter baumannii (A. baumannii)* strains, imipenem and piperacillin-tazobactam (PTZ) resistant *Pseudomonas aeruginosa (P. aeruginosa)* strains.The choice to include *E. coli* and several bacterial species from the ESKAPE group in our study was primarily based on the predominance of these bacterial species in the samples taken from our hospital. Additionally, in recent times, *E. coli* strains have become increasingly resistant.

Information on the number of hospitalization days per year was obtained from the hospital administration. These administrative data detail the activity of all clinical departments in terms of annual hospitalization days.

Data on antibiotic consumption across all care departments were collected using STKMED^®^ software utilized by the hospital pharmacy. This software provides raw data on antibiotic consumption in terms of units expressed in grams or international units. These data were then converted into the number of Defined Daily Doses (DDDs) [61] using EXCEL^®^ software to calculate the international consumption indicator expressed in DDDs per 1000 hospitalization days as adopted by the WHO guidelines [29,62].

For calculating the number of Defined Daily Doses (DDD), proceed as follows: [62]

First, calculate the total quantity in grams (g) of the antibacterial used per year:Quantity (g) = annual consumption quantity × dosage (g) 
where
annual consumption quantity (units) = Number of boxes per year × content

Then, according to the WHO’s DDD table, the formula for the number of DDDs is as follows:Number of DDDs = Quantity (g)/DDD of the active ingredient according to WH 

The formula to calculate the International Consumption Indicator according to the WHO guidelines [61] is as follows:International Consumption Indicator = (Number of DDD/Number of Hospitalization Days) × 1000

The collection of bacteriological data necessary for this study was carried out using the SIRSCAN^®^ software from the microbiology laboratory at Charles Nicolle Hospital in Tunis. Based on the collected data, we calculated the percentage of resistance and then the resistance density.
Percentage of Resistance = [(Number of tested strains − Number of sensitive strains)/Number of tested strains] × 100
where:Number of resistant strains = Number of tested strains − Number of sensitive strains
Incidence Density of Resistance = (Number of resistant strains/Number of Hospitalization Days) × 1000

Statistical analyses of data related to antibiotic consumption and bacterial resistance density were performed using SPSS^®^ software version 23. It has been demonstrated that using the incidence density of resistance yields statistically more reliable results compared to using the percentage of bacterial strain resistance [63].

The normality of the variables studied was assessed in SPSS^®^ using the Shapiro-Wilk test [64,65]. A variable is considered to have a normal distribution if *p* > 0.05.

The pharmaco-epidemiological analysis translating the correlation, whether direct or indirect, between antibiotic consumption and the emergence of bacterial resistance, was carried out using Pearson’s R parametric test if the distribution of at least one of the two variables tested together is normal, or Spearman’s rho nonparametric test if neither variable has a normal distribution [66]. Correlations are significant if their *p*-value ≤ 0.05.

To evaluate trends, whether upward or downward, linear regressions were tested using SPSS^®^ software. These values are significant and retained if their *p*-value ≤ 0.05.

β represents the slope of the linear regression curve depicting the relationship between the independent variable (in our case, the years) and the dependent variable (in our case, bacterial resistance or the consumption of an antibiotic). This means that the slope of an independent variable reflects the expected change in the dependent variable. If the value of this slope is positive, it indicates that the relationship between the two variables is positive and they evolve in the same direction (in our case, this means that over the years, consumption increases or resistance increases). If the value of the slope is negative, then the relationship between the two variables is negative, and the two values evolve in opposite directions (therefore, in our case, from 2010 to 2022, the consumption of the antibiotic in question or the resistance of the germ has a downward trend).

There was no ethical approval for this study as no patients were involved.

## 5. Conclusions

The emergence of bacterial resistance is now one of the main challenges in the healthcare field of the 21st century. Addressing this alarming issue requires the implementation of strict measures through national and international action plans to rationalize the use of antibiotics, the primary factor promoting antimicrobial resistance.

The main objective of this work was to study the possible correlation between the consumption of various antibiotics and the increase in resistance density of the studied bacteria. This study revealed a significant decrease in overall antibiotic consumption between 2010 and 2022 by −55.04%, demonstrating the positive impact of actions taken by hospital pharmacists in collaboration with prescribers to rationalize antibiotic use. However, the “Watch” group accounted for almost half of the antibiotic consumption, which is quite high compared to WHO recommendations.

Regarding bacterial resistance to antibiotics, the results revealed certain significant upward trends, such as *E. coli* resistance to 3GCs and carbapenems, and *K. pneumoniae* resistance to carbapenems.

More than seventy correlations were recorded between antibiotic consumption and bacterial resistance density. Among the most relevant, we note the correlations between carbapenem consumption and the increased resistance density of *E. coli* to 3GCs and carbapenems, and of *K. pneumoniae* to carbapenems. This study revealed significant positive correlations between the increased consumption of PTZ and the emergence of several bacterial resistances, contrary to the findings in the literature.

The perspectives of this study at the scale of our hospital aim to reduce the consumption of antibiotics from the “Watch” group, particularly carbapenems, by carefully evaluating their prescriptions. Additionally, it is crucial to shorten the duration of C3G use by re-evaluating treatments after 72 h and promoting therapeutic de-escalation to antibiotics from the “Access” group as soon as possible. Strict monitoring of prescriptions for antibiotics from the “Reserve” group is essential to preserve their effectiveness.

The results of this study will be discussed point by point during the antimicrobial stewardship committee meeting to outline guidelines aimed at reducing antimicrobial resistance.

## Figures and Tables

**Figure 1 antibiotics-14-00135-f001:**
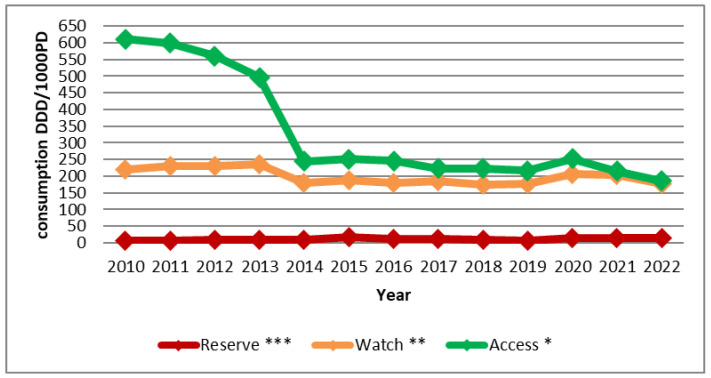
The consumption of the three groups in the “AWaRe” classification from 2010 to 2022 in defined daily doses per 1000 patient days. * Access: This group includes antibiotics recommended as first- and second-line treatments in protocols for common infections. (Example: Amoxicillin). ** Watch: This group also includes first- and second-line antibiotics for specific infections rather than common ones. Since the drugs in this group have a higher potential for developing resistance, they must be an integral part of antimicrobial stewardship programs ASP (Example: Cefotaxim). *** Reserve: This group includes last-resort antibiotics used when the patient’s life is at risk due to an infection caused by multidrug-resistant bacteria (Example: Tigecycline). DDD/1000PD: defined daily doses per 1000 patient days.

**Figure 2 antibiotics-14-00135-f002:**
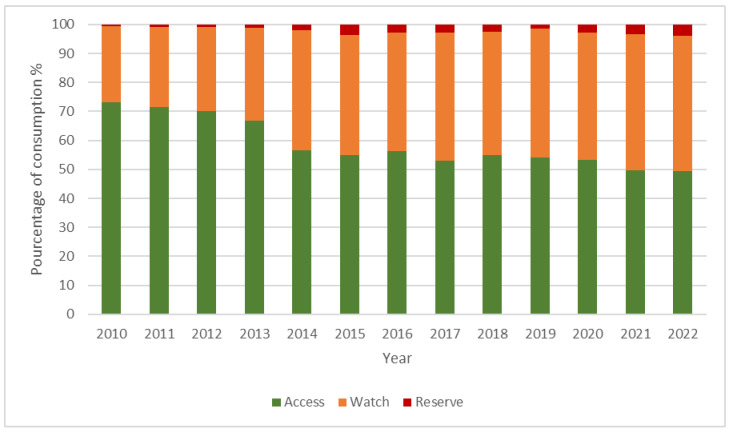
Share of “Access”, “Watch”, and “Reserve” groups in antibiotic consumption between 2010 and 2022.

**Table 1 antibiotics-14-00135-t001:** Results of linear regressions of antibiotic consumption between 2010 and 2022.

Antibiotics	Linear Regression		
	β	P	Trend *
**Benzylpenicillin**	**−0.908**	**0.000**	**-**
**Benzathine- Benzylpenicillin**	**−0.932**	**0.000**	**-**
**Oxacillin**	**−0.837**	**0.000**	**-**
**Ampicillin**	**−0.181**	**0.553**	**Nonsignificant**
**Amoxicillin**	**−0.875**	**0.000**	**-**
**Amoxicillin-clavulanic acid**	**−0.494**	**0.086**	**Nonsignificant**
**Piperacillin**	**−0.687**	**0.009**	**-**
**Piperacillin + Tazobactam**	**0.861**	**0.000**	**+**
**Penicillins**	**−0.865**	**0.000**	**-**
**Cefazolin**	**0.023**	**0.940**	**Nonsignificant**
**Cefuroxime sodium**	**−0.708**	**0.007**	**-**
**Cefotaxime**	**0.708**	**0.007**	**+**
**Ceftazidime**	**−0.538**	**0.058**	**Nonsignificant**
**Ceftriaxone**	**0.536**	**0.059**	**Nonsignificant**
**Third-generation cephalosporins**	**0.718**	**0.006**	**+**
**Cephalosporins**	**0.703**	**0.007**	**+**
**Imipenem-cilastatin**	**0.873**	**0.000**	**+**
**Meropenem**	**0.463**	**0.111**	**Nonsignificant**
**Ertapenem**	**−0.803**	**0.001**	**-**
**Carbapenems**	**0.808**	**0.001**	**+**
**Beta-lactams**	**−0.833**	**0.000**	**-**
**Amikacin**	**0.562**	**0.046**	**+**
**Gentamicin**	**0.425**	**0.148**	**Nonsignificant**
**Aminoglycosides**	**0.500**	**0.082**	**Nonsignificant**
**Vancomycin**	**0.526**	**0.065**	**Nonsignificant**
**Teicoplanin**	**0.565**	**0.044**	**+**
**Glycopeptides**	**0.750**	**0.003**	**+**
**Ofloxacin**	**−0.804**	**0.001**	**-**
**Ciprofloxacin**	**−0.895**	**0.000**	**-**
**Levofloxacin**	**0.118**	**0.700**	**Nonsignificant**
**Fluoroquinolones**	**−0.905**	**0.000**	**-**
**Erythromycin**	**−0.845**	**0.000**	**-**
**Spiramycin**	**−0.857**	**0.000**	**-**
**Clindamycin**	**0.731**	**0.005**	**+**
**Pristinamycin**	**−0.873**	**0.000**	**-**
**Macrolides**	**−0.902**	**0.000**	**-**
**Tigecycline**	**0.891**	**0.000**	**+**
**Trimethoprim-sulfamethoxazole (TMP-SMX)**	**−0.752**	**0.003**	**-**
**Rifampicin**	**−0.555**	**0.049**	**-**
**Fusidic acid**	**−0.823**	**0.001**	**-**
**Fosfomycin**	**−0.491**	**0.088**	**Nonsignificant**
**Linezolid**	**0.520**	**0.068**	**Nonsignificant**
**Colistin**	**0.406**	**0.169**	**Nonsignificant**
**Metronidazole**	**−0.182**	**0.551**	**Nonsignificant**

* (**-**) Significant negative trend; (**+**) significant positive trend.

**Table 2 antibiotics-14-00135-t002:** Results of linear regressions of bacterial strain resistance densities between 2010 and 2022.

Bacterial Strains	Linear Regression		
	β	P	Trend *
*E. coli* resistant to third-generation cephalosporins	0.585	0.036	+
*E. coli* resistant to carbapenems	0.755	0.003	+
ESBL-producing *E. coli*	0.627	0.022	+
*K. pneumoniae* resistant to third-generation cephalosporins	0.480	0.097	Nonsignificant
*K. pneumoniae* resistant to carbapenems	0.680	0.028	+
*P. aeruginosa* resistant to imipenem	−0.358	0.230	Nonsignificant
*P. aeruginosa* resistant to piperacillin-tazobactam	0.056	0.855	Nonsignificant
*A. baumannii* resistant to imipenem	0.551	0.051	Nonsignificant
Methicillin-resistant *Staphylococcus aureus*	0.506	0.078	Nonsignificant

*P.: Pseudomonas; K.: Klebsiella; A.: Acinetobacter; E.: Esherichia*; ESBL: Extended-Spectrum Beta-Lactamase. * (+) significant positive trend.

**Table 3 antibiotics-14-00135-t003:** Correlation between the increase in resistance density of *Escherichia coli* strains to third-generation cephalosporins and antibiotic consumption.

Increase in Resistance Density of *Escherichia coli* Strains to Third-Generation Cephalosporins				
Antibiotics	Bivariate Correlation			
	R^2^	P	Direction of Correlation	Nature of Correlation *
Amoxicillin	−0.611	0.027	Decreasing	Indirect
Piperacillin + Tazobactam	0.611	0.027	Increasing	Indirect
Cefotaxime	0.584	0.036	Decreasing	Direct
Third-generation cephalosporins	0.578	0.039	Increasing	Direct
Imipenem + Cilastatine	0.684	0.010	Increasing	Indirect
Carbapenems	0.693	0.009	Increasing	Indirect
Aminoglycosides	0.583	0.036	Undefined direction	Indirect
Ofloxacin	−0.555	0.049	Decreasing	Indirect
Ciprofloxacin	−0.589	0.034	Decreasing	Indirect
**Fluoroquinolones**	−0.677	0.011	Decreasing	Indirect
Trimethoprim-sulfamethoxazole	−0.633	0.020	Decreasing	Indirect

***** There are two types of correlations. Direct correlations occur when bacterial resistance to an antibiotic is correlated with the consumption of that same drug. Indirect correlations occur when the consumption of one antibiotic causes bacterial resistance to another antibiotic.

**Table 4 antibiotics-14-00135-t004:** Correlation between the increase in resistance density of Extended-Spectrum Beta-Lactamase (ESBL) producing *Escherichia coli* strains and antibiotic consumption.

Increase in Resistance Density of ESBL-Producing *Escherichia coli* Strains				
Antibiotics	Bivariate Correlation			
	R^2^	P	Direction of Correlation	Nature of Correlation *
Oxacillin	**−0.878**	**0.000**	Decreasing	**Indirect**
Amoxicillin	**−0.816**	**0.001**	Decreasing	**Indirect**
Imipenem + Cilastatine	**0.789**	**0.001**	Increasing	**Indirect**
**Carbapenems**	**0.861**	**0.000**	Increasing	**Indirect**
Ciprofloxacin	**−0.848**	**0.000**	Decreasing	**Indirect**
**Fluoroquinolones**	**−0.814**	**0.001**	Decreasing	**Indirect**
Vancomycin	**0.807**	**0.001**	Undefined direction	**Indirect**
**Glycopeptides**	**0.634**	**0.020**	Increasing	**Indirect**
Trimethoprim-sulfamethoxazole	**−0.874**	**0.000**	Decreasing	**Indirect**

***** There are two types of correlations. Direct correlations occur when bacterial resistance to an antibiotic is correlated with the consumption of that same drug. Indirect correlations occur when the consumption of one antibiotic causes bacterial resistance to another antibiotic. ESBL: Extended-Spectrum Beta-Lactamase.

**Table 5 antibiotics-14-00135-t005:** Correlation between the increase in resistance density of carbapenem-resistant *Escherichia coli* strains and antibiotic consumption.

Increase in Resistance Density of *Escherichia coli* Strains to Carbapenems				
Antibiotics	Bivariate Correlation			
	R^2^	P	Direction of Correlation	Nature of Correlation *
Oxacillin	**−0.570**	**0.042**	**Decreasing**	**Indirect**
Amoxicillin	**−0.697**	**0.008**	**Decreasing**	**Indirect**
Piperacillin + Tazobactam	**0.768**	**0.002**	**Increasing**	**Indirect**
Cefotaxime	**0.725**	**0.005**	Increasing	**Indirect**
**Third-generation cephalosporin**	**0.719**	**0.006**	**Increasing**	**Indirect**
Imipenem + Cilastatine	**0.680**	**0.011**	**Increasing**	**Direct**
Ertapenem	**−0.621**	**0.024**	**Decreasing**	**Direct**
**Carbapenems**	**0.626**	**0.022**	**Increasing**	**Direct**
Ofloxacin	**−0.597**	**0.031**	**Decreasing**	**Indirect**
Ciprofloxacin	**−0.694**	**0.008**	**Decreasing**	**Indirect**
Fluoroquinolones	**−0.796**	**0.001**	**Decreasing**	**Indirect**
Teicoplanin	**0.556**	**0.048**	**Increasing**	**Indirect**

***** There are two types of correlations. Direct correlations occur when bacterial resistance to an antibiotic is correlated with the consumption of that same drug. Indirect correlations occur when the consumption of one antibiotic causes bacterial resistance to another antibiotic.

**Table 6 antibiotics-14-00135-t006:** Correlation between the increase in resistance density of *Klebsiella pneumoniae* strains to third-generation cephalosporins and antibiotic consumption.

Increase in Resistance Density of *Klebsiella pneumoniae* Strains to Third-Generation Cephalosporins				
Antibiotics	Bivariate Correlation			
	R^2^	P	Direction of Correlation	Nature of Correlation *
Piperacillin	**−0.647**	**0.017**	**Decreasing**	**Indirect**
Piperacillin + Tazobactam	**0.707**	**0.007**	**Increasing**	**Indirect**
Cefotaxime	**0.615**	**0.025**	**Increasing**	**Direct**
**Third-generation cephalosporins**	**0.669**	**0.012**	**Increasing**	**Direct**
Amikacin	**0.639**	**0.019**	**Increasing**	**Indirect**
Rifampicin	**−0.629**	**0.021**	**Decreasing**	**Indirect**

***** There are two types of correlations. Direct correlations occur when bacterial resistance to an antibiotic is correlated with the consumption of that same drug. Indirect correlations occur when the consumption of one antibiotic causes bacterial resistance to another antibiotic.

**Table 7 antibiotics-14-00135-t007:** Correlation between the increase in resistance density of carbapenem-resistant *Klebsiella pneumoniae* strains and antibiotic consumption.

Increase in Resistance Density of Carbapenem-Resistant *Klebsiella pneumoniae* Strains				
Antibiotics	Bivariate Correlation			
	R^2^	P	Direction of Correlation	Nature of Correlation *
Oxacillin	**−0.567**	**0.043**	**Decreasing**	**Indirect**
Amoxicillin	**−0.787**	**0.001**	**Decreasing**	**Indirect**
Piperacillin	**−0.616**	**0.025**	**Decreasing**	**Indirect**
Piperacillin + Tazobactam	**0.707**	**0.007**	**Increasing**	**Indirect**
Cefotaxime	**0.671**	**0.012**	**Increasing**	**Indirect**
**Third-generation cephalosporins**	**0.908**	**0.000**	**Increasing**	**Indirect**
Imipenem + Cilastatine	**0.559**	**0.047**	**Increasing**	**Direct**
Amikacin	**0.688**	**0.009**	**Increasing**	**Indirect**
Ciprofloxacin	**−0.836**	**0.000**	**Decreasing**	**Indirect**
**Fluoroquinolones**	**−0.891**	**0.000**	**Decreasing**	**Indirect**
Teicoplanin	**0.726**	**0.005**	**Increasing**	**Indirect**
Rifampicin	**−0.629**	**0.021**	**Decreasing**	**Indirect**

***** There are two types of correlations. Direct correlations occur when bacterial resistance to an antibiotic is correlated with the consumption of that same drug. Indirect correlations occur when the consumption of one antibiotic causes bacterial resistance to another antibiotic.

**Table 8 antibiotics-14-00135-t008:** Correlation between the increase in resistance density of *Pseudomonas aeruginosa* strains to piperacillin-tazobactam and antibiotic consumption.

Increase in Resistance Density of *Pseudomonas aeruginosa* Strains to Piperacillin-Tazobactam				
Antibiotics	Bivariate Correlation			
	R^2^	P	Direction of Correlation	Nature of Correlation *
Ampicillin	0.613	0.026	Undefined direction	Indirect
Gentamicin	0.806	0.001	Undefined direction	Indirect
Aminoglycosides	0.752	0.003	Undefined direction	Indirect

***** There are two types of correlations. Direct correlations occur when bacterial resistance to an antibiotic is correlated with the consumption of that same drug. Indirect correlations occur when the consumption of one antibiotic causes bacterial resistance to another antibiotic.

**Table 9 antibiotics-14-00135-t009:** Correlation between the increase in resistance density of Acinetobacter baumannii strains to imipenem and antibiotic consumption.

Increase in Resistance Density of *Acinetobacter baumannii* Strains to Imipenem				
Antibiotics	Bivariate Correlation			
	R^2^	P	Direction of Correlation	Nature of Correlation *
Amoxicillin	**−0.641**	**0.018**	**Decreasing**	**Indirect**
Piperacillin + Tazobactam	**0.706**	**0.007**	**Increasing**	**Indirect**
Cefotaxime	**0.703**	**0.007**	**Increasing**	**Indirect**
**Third-generation cephalosporins**	**0.652**	**0.016**	**Increasing**	**Indirect**
Imipenem + Cilastatine	**0.555**	**0.049**	**Increasing**	**Direct**
Ciprofloxacin	**−0.635**	**0.020**	**Decreasing**	**Indirect**
**Fluoroquinolones**	**−0.558**	**0.047**	**Decreasing**	**Indirect**
Rifampicin	**−0.691**	**0.009**	**Decreasing**	**Indirect**

***** There are two types of correlations. Direct correlations occur when bacterial resistance to an antibiotic is correlated with the consumption of that same drug. Indirect correlations occur when the consumption of one antibiotic causes bacterial resistance to another antibiotic.

**Table 10 antibiotics-14-00135-t010:** Correlation between the increase in resistance density of Methicillin-resistant *Staphylococcus aureus* (MRSA) strains and antibiotic consumption.

Increase in Resistance Density of Methicillin-Resistant *Staphylococcus aureus* (MRSA) Strains				
Antibiotics	Bivariate Correlation			
	R^2^	P	Direction of Correlation	Nature of Correlation *
Oxacillin	**−0.559**	**0.047**	**Decreasing**	**Direct**
Piperacillin + Tazobactam	**0.591**	**0.033**	**Increasing**	**Indirect**
Ceftriaxone	**0.553**	**0.050**	**Increasing**	**Indirect**
Third-generation cephalosporins	**0.592**	**0.033**	**Increasing**	**Indirect**
Imipenem + Cilastatine	**0.616**	**0.025**	**Increasing**	**Indirect**
Carbapenemes	**0.644**	**0.017**	**Increasing**	**Indirect**
Amikacin	**0.589**	**0.034**	**Increasing**	**Indirect**
Gentamicin	**0.581**	**0.037**	**Undefined direction**	**Indirect**
Aminoglycosides	**0.630**	**0.021**	**Undefined direction**	**Indirect**
Spiramycin	**−0.566**	**0.044**	**Decreasing**	**Indirect**
Ciprofloxacin	**−0.575**	**0.040**	**Decreasing**	**Indirect**
Fusidic acid	**−0.577**	**0.039**	**Decreasing**	**Indirect**
Rifampicin	**−0.572**	**0.041**	**Decreasing**	**Indirect**

***** There are two types of correlations. Direct correlations occur when bacterial resistance to an antibiotic is correlated with the consumption of that same drug. Indirect correlations occur when the consumption of one antibiotic causes bacterial resistance to another antibiotic.

## Data Availability

The data were collected from software installed at the hospital's pharmacy and microbiology laboratory; they are not available on any website.

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
