# Peer review of "Pharmaco-Epidemiological Study and Correlation Between Antibiotic Resistance and Antibiotic Consumption in a Tunisian Teaching Hospital from 2010 to 2022"

_antibiotics, 2025, doi:10.3390/antibiotics14020135_

Round 1
Reviewer 1 Report
Comments and Suggestions for Authors
The study from Charles Nicolle Hospital in Tunis, Tunisia, examines the relationship between antibiotic consumption and bacterial resistance from 2010 to 2022. Despite a 55.04% reduction in antibiotic use, "Watch" group antibiotics still constituted nearly half of the total consumption, surpassing WHO recommendations. Significant correlations were found between antibiotic use and resistance, notably increasing E. coli resistance to third-generation cephalosporins and carbapenems, and rising K. pneumoniae resistance to carbapenems. The findings underline the need for enhanced antibiotic stewardship and comprehensive action plans to combat antimicrobial resistance.
Minor issues
1. What is “AWaRe” classification? What are “Access”, “Watch” and “Reserve” groups? The authors should briefly introduce or define these abbreviations.
2. line 84: What is the “β”? What does this value mean?
Author Response
Comments 1: What is “AWaRe” classification? What are “Access”, “Watch” and “Reserve” groups? The authors should briefly introduce or define these abbreviations.
Response1: Thank you for pointing this out. We want to specify that we mentioned Within the Materials and Methods section, page 16 from line 411 to line 428, the definition of the WHO AWaRe classification with its three groups. Additionally, we included in Annex 1 page 23, a table classifying the antibiotics available in our hospital according to their ATC code and their classification into one of the 3 AWaRe groups and, below Figure 1 on page 2, we added the definition of the three groups.
Comments 2: line 84: What is the “β”? What does this value mean?
Response 2: β represents the slope of the linear regression curve depicting the relationship between the independent variable (in our case, the years) and the dependent variable (in our case, bacterial resistance or the consumption of an antibiotic). This means that the slope of an independent variable reflects the expected change in the dependent variable. If the value of this slope is positive, it indicates that the relationship between the two variables is positive and they evolve in the same direction (in our case, this means that over the years, consumption increases or resistance increases). If the value of the slope is negative, then the relationship between the two variables is negative and the two values evolve in opposite directions (therefore, in our case, from 2010 to 2022, the consumption of the antibiotic in question or the resistance of the germ has a downward trend).
We added this explication in Materials and Methods section page 18 from line 489 to 498.

Reviewer 2 Report
Comments and Suggestions for Authors
Kasbi et al have worked on reviewing the antibiotic consumption data available at a Tunisian teaching hospital over a period of 12 years and co-relate it with antimicrobial resistance. The authors have done a nice work on compiling information and identifying trends in different categories.
While the publication is well written, a few more details need to be added to highlight the effect of implementing strict measures of regulating antibiotic use to reduce the emergence of more drug resistant strains.
Comments -
Page 2 Line 60 - Please introduce the AWare classification system in a couple of sentences. The authors do talk about the Access, Reserve, and Watch groups in the methods section but please add a couple sentences in this section as well
Page 2 Line 62 - do the authors mean to use "recorded" instead of "recorder"
Page 4 Line 96 - do the authors mean to use "recorded" instead of "recorder"
Page 6 Line 153 - Could the authors please quantify "significant" as it is a relative term and doesn't differentiate between 2-fold vs 10-fold
Page 11 Line 213-220 - The authors did a nice job including the effect of COVID on the consumption of antibiotics.
Page 11 Line 235-236 - Could the authors please add a few sentences to discuss the "Watch" group. Please mention the WHO recommendation for "Watch" group. When did the Watch group consumption reach 60% (as mentioned in Line 236) if it is 46.89% in 2022.
Page 11 Line 239 - Do the "Access" and "Watch" groups use a very specific set of medications that are unique to each group? "Watch" group "specific infections" need to b explained better
Page 14 Line 366 - Please give more details about the classification in a couple of sentences
Page 14 Line 370 - As mentioned above, "specific infections" need to be explained in more details
Page 14 Line 380-385 - Why did the authors go with this select of bacteria and not the complete profile of ESKAPE pathogens? Is there a reason the authors did not look to classify in sets of gram-positive and gram-negative bacteria
Page 16 Line 430-433 - Did the authors see an effect of other drugs for health conditions such as cardiac disease, diabetes, arthiritis etc on antibiotic prescription and co-relation with increase in resistance to antibacterials
Comments on the Quality of English LanguageThere seem to be a couple of typos that are mentioned in the comments above
Author Response
Comments 1: Page 2 Line 60 - Please introduce the AWare classification system in a couple of sentences. The authors do talk about the Access, Reserve, and Watch groups in the methods section but please add a couple sentences in this section as well.
Response1: Thank you for pointing this out. We want to specify that we mentioned Within the Materials and Methods section, page 16 from line 411 to line 428, the definition of the WHO AWaRe classification with its three groups. Additionally, we included in Annex 1 page 23, a table classifying the antibiotics available in our hospital according to their ATC code and their classification into one of the 3 AWaRe groups and, below Figure 1 on page 2, we added the definition of the three groups.
We also added references :
reference 57: 2021 AWaRe classification
And the reference 59: World Health Organization. The WHO AWaRe (Access, Watch, Reserve) antibiotic book. Geneva: WHO; 2022.
Comments 2: Page 2 Line 62 - do the authors mean to use "recorded" instead of "recorder"
Response2: We agree. Accordingly, we have made this rectification
Comments 3: Page 4 Line 96 - do the authors mean to use "recorded" instead of "recorder"
Response3: We agree. Accordingly, we have made this rectification
Comments 4: Page 6 Line 153 - Could the authors please quantify "significant" as it is a relative term and doesn't differentiate between 2-fold vs 10-fold
Response 4: We agree. We added in Page 7 Line 158, Linear regressions are significant only if the p-value is ⩽0.05.
Comments 5: Page 11 Line 213-220 - The authors did a nice job including the effect of COVID on the consumption of antibiotics.
Response 5: Thank you so much
Comments 6: Page 11 Line 235-236 - Could the authors please add a few sentences to discuss the "Watch" group. Please mention the WHO recommendation for "Watch" group. When did the Watch group consumption reach 60% (as mentioned in Line 236) if it is 46.89% in 2022.
Response 6: As mentioned in response 1, we have added some explanations about the AWaRe classification. However, the WHO recommendations have required a minimum percentage of 60% for antibiotics from the Access group, but they have not proposed a specific percentage to be adhered to for antibiotics from the Watch or Reserve groups.
Comments 7: Page 11 Line 239 - Do the "Access" and "Watch" groups use a very specific set of medications that are unique to each group? "Watch" group "specific infections" need to b explained better
Response 7: Thank you for pointing this out. We want to clarify that each group of the AWaRe classification encompasses a list of antibiotics that share common properties (Annex 1 clarifies this classification). Regarding specific infections for which antibiotics from the Watch group are indicated, the WHO has published a book specifying these infections. We have added the reference to this book (reference 59).
Comments 8: Page 14 Line 366 - Please give more details about the classification in a couple of sentences
Response 8: We have added some clarifications about the classification as mentioned in response 1
Comments 9: Page 14 Line 370 - As mentioned above, "specific infections" need to be explained in more details
Response 9: As mentioned in Response 7, Regarding specific infections for which antibiotics from the Watch group are indicated, the WHO has published a book specifying these infections. We have added the reference to this book (reference 59).
Comments 10: Page 14 Line 380-385 - Why did the authors go with this select of bacteria and not the complete profile of ESKAPE pathogens? Is there a reason the authors did not look to classify in sets of gram-positive and gram-negative bacteria
Response10: The choice to include E. coli and several bacterial species from the ESKAPE group in our study was primarily based on the predominance of these bacterial species in the samples taken from our hospital. Additionally, in recent times, E. coli strains have become increasingly resistant. We added this explication in Page 16 line 442.
Comments 11: Page 16 Line 430-433 - Did the authors see an effect of other drugs for health conditions such as cardiac disease, diabetes, arthiritis etc on antibiotic prescription and co-relation with increase in resistance to antibacterials
Response11: Our study did not take into account the impact of other classes of drugs on antibiotic prescription and co-relation with increase in resistance to antibacterials; we focused solely on the impact of antibiotic consumption. However, we thank you for this very interesting topic, which could be a subject for future work.

Reviewer 3 Report
Comments and Suggestions for Authors
The present retrospective study highlights a very important issue i.e correlating antibiotic resistance with consumption. There are few issues which need to be addressed:
1. It is stated that ethical approval was not taken as the study does not involve patients. However, this is against the publication requirements as any research involving patients or data has to be conducted after obtaining ethical approval from the respective Institute ethics committee. Was the study granted waiver from Ethics committee? If yes, it has to be stated accordingly.
2. In this kind of retrospective data-based study, approval from hospital’s administrative authority needs to be taken which I assume has not been taken by the investigators.
3. Methods: Few areas need to be explained in detail to improve the robustness of the study:
· Give a brief description of the study setting viz. type of hospital, patient load etc.
· How was the incidence density of resistance calculated?
· Please explain the STK MED software used for antibiotic consumption determination? What are the units used for reporting the consumption estimates in the software used? How similar or different is the software from WHO Antimicrobial Consumption (AMC) tool?
· Please describe the type of data collected for example systemic or topical preparations.
4. RESULTS:
· There was a steep decline in “access” group consumption from 2013 to 2014. Overall, we could witness decrease in the percentage of antibiotic consumption from access group over years, which is in fact not favourable. Could this be explained in the current study?
· Data in table in figure 2 depicts percentages of consumption; commas need to be replaced with decimal points even for data in various tables.
· The authors are requested to present the antibiotics as WHO ATC/DDD codes.
· Table 3 title mentions only 3rd generation cephalosporins while the table includes data from other groups of antibiotics also; please correct.
· What is the basis for classifying the nature of correlation into direct and indirect?
5. The study limitations mainly being single-centric and limited generalizability, retrospective data etc. need to be discussed.
6. Conclusion: The authors have discussed many correlations between resistance and consumption; there is need to mention some specific recommendations depending upon the study findings under conclusion section.
Comments on the Quality of English Language
Moderate language editing required.
Author Response
Comments 1: It is stated that ethical approval was not taken as the study does not involve patients. However, this is against the publication requirements as any research involving patients or data has to be conducted after obtaining ethical approval from the respective Institute ethics committee. Was the study granted waiver from Ethics committee? If yes, it has to be stated accordingly.
Response1: We would like to clarify that the retrospective data used in our study do not disclose any personal information about patients. Our analysis includes only aggregate data on overall antibiotic consumption and germ resistance, sourced from the SIRSCAN software available in the hospital’s microbiology laboratory. These data are anonymized and do not contain any identifiable patient information. Furthermore, the bacterial resistance data collected in our hospital are part of the Tunisian Antimicrobial Resistance Network (LART), which includes contributions from seven other data collection sites across Tunisia. This network publishes aggregated data, ensuring that individual patient privacy is maintained. Given the nature of our study, which relies solely on anonymized, aggregated data, it does not fall under the category of research requiring approval from the hospital's ethics committee. Would you like any further adjustments?
Comments 2: In this kind of retrospective data-based study, approval from hospital’s administrative authority needs to be taken which I assume has not been taken by the investigators.
Response2: We want to clarify that the data were collected with the prior approval from the Nosocomial Infection Control Committee (CLIN) of our hospital, which has the administrative authority to manage this data.
Comments 3: Give a brief description of the study setting viz. type of hospital, patient load etc.
Response3: We agree. We added in page 16 Line 404 few sentences describing our hospital.
Comments 4: How was the incidence density of resistance calculated?
Response 4: We added in page 17 line 474, the formula for calculating the incidence density of bacterial resistance. Additionally, we have included all formulas used in this study starting from page 17, line 455.
Comments 5: Please explain the STK MED software used for antibiotic consumption determination? What are the units used for reporting the consumption estimates in the software used? How similar or different is the software from WHO Antimicrobial Consumption (AMC) tool?
Response5: We agree. The STKMED® software utilized by the hospital pharmacy provided raw data on antibiotic consumption in units expressed in grams or international units. Efforts were made to calculate the number of DDDs and then the international indicator DDD/1000 PD to align with WHO recommendations. The results found were double-checked.
Comments 6: Please describe the type of data collected for example systemic or topical preparations.
Response6: We would like to inform you that the data on the included antibiotics have been detailed in the Materials and Methods section page 16 line 410.
Comments 7: There was a steep decline in “access” group consumption from 2013 to 2014. Overall, we could witness decrease in the percentage of antibiotic consumption from access group over years, which is in fact not favourable. Could this be explained in the current study?
Response7: The decrease in the consumption of antibiotics from the Access group between 2013 and 2014 is probably due to the emergence of bacterial resistance to these antibiotics, rendering them ineffective. This has led to their replacement by antibiotics from the Watch group. For example, urinary infections, previously treated with amoxicillin, have become resistant to this molecule
Comments8: Data in table in figure 2 depicts percentages of consumption; commas need to be replaced with decimal points even for data in various tables.
Response8: We agree. Accordingly, we have made this rectification
Comments 9: The authors are requested to present the antibiotics as WHO ATC/DDD codes.
Response9: We agree. We added in Annex 1 a table classifying the antibiotics available in our hospital according to their ATC code and their classification into one of the 3 AWaRe groups.
Comments 10: Table 3 title mentions only 3rd generation cephalosporins while the table includes data from other groups of antibiotics also; please correct.
Response10: We would like to explain that Table 3 examines the influence of the consumption of different antibiotics on the occurrence of C3G-resistant E. coli strains.
Comments 11: What is the basis for classifying the nature of correlation into direct and indirect?
Response11: There are two types of correlations.
- Direct correlations occur when bacterial resistance to an antibiotic is correlated with the consumption of that same drug.
- Indirect correlations occur when the consumption of one antibiotic causes bacterial resistance to another antibiotic.
We have added this explanation below the tables that reveal these correlations.
Comments 12: The study limitations mainly being single-centric and limited generalizability, retrospective data etc. need to be discussed.
Response12: We agree. We added in page 15 line 387 two paragraphs highlighting the strengths and limitations of this study.
Comments13: Conclusion: The authors have discussed many correlations between resistance and consumption; there is need to mention some specific recommendations depending upon the study findings under conclusion section.
Response13: We agree. We added in page 18 line 521 few sentences explaining the perspectives of this study at the scale of our hospital.

Round 2
Reviewer 2 Report
Comments and Suggestions for Authors
I am satisfied with the authors' edits and remarks
Reviewer 3 Report
Comments and Suggestions for Authors
No further comments from my side.